# Salivary Protein Cyclin-Dependent Kinase-like from Grain Aphid *Sitobion avenae* Suppresses Wheat Defense Response and Enhances Aphid Adaptation

**DOI:** 10.3390/ijms25094579

**Published:** 2024-04-23

**Authors:** Yumeng Zhang, Xiaobei Liu, Yu Sun, Yong Liu, Yong Zhang, Tianbo Ding, Julian Chen

**Affiliations:** 1College of Plant Health and Medicine, Qingdao Agricultural University, Qingdao 266109, China; zhangyumemg1017@163.com; 2State Key Laboratory for Biology of Plant Diseases and Insect Pests, Institute of Plant Protection, Chinese Academy of Agricultural Sciences, Beijing 100193, China; xiaobeiliu7@163.com (X.L.); yu1542286707@126.com (Y.S.);; 3College of Plant Protection, Shandong Agricultural University, Taian 271018, China; liuyong@sdau.edu.cn

**Keywords:** *Sitobion avenae*, salivary protein, defense response, aphid adaptation, bacterial type III secretion system

## Abstract

Aphids are insect pests that suck phloem sap and introduce salivary proteins into plant tissues through saliva secretion. The effector of salivary proteins plays a key role in the modulation of host plant defense responses and enhancing aphid host adaptation. Based on previous transcriptome sequencing results, a candidate effector cyclin-dependent kinase-like (CDK) was identified from the grain aphid *Sitobion avenae*. In this study, the function of SaCDK in wheat defense response and the adaptation of *S. avenae* was investigated. Our results showed that the transient overexpression of SaCDK in tobacco *Nicotiana benthamiana* suppressed cell death triggered by mouse pro-apoptotic protein-BAX or *Phytophthora infestans* PAMP-INF1. SaCDK, delivered into wheat cells through a *Pseudomonas fluorescens*-mediated bacterial type III secretion system, suppressed callose deposition in wheat seedlings, and the overexpression of SaCDK in wheat significantly decreased the expression levels of salicylic acid and jasmonic acid signaling pathway-related genes phenylalanine ammonia lyase (*PAL*), pathogenesis-related 1 protein (*PR1*), lipoxygenase (*LOX*) and Ω-3 fatty acid desaturase (*FAD*). In addition, aphid bioassay results showed that the survival and fecundity of *S. avenae* were significantly increased while feeding on the wheat plants carrying SaCDK. Taken together, our findings demonstrate that the salivary protein SaCDK is involved in inhibiting host defense response and improving its host adaptation, which lays the foundation to uncover the mechanism of the interaction of cereal aphids and host plants.

## 1. Introduction

In nature, plants are attacked by various biotic factors such as pathogens and insect pests. In response to these biotic stresses, plants trigger a series of defense responses through various signaling pathways [1]. Phytohormones can act as important molecules involved in defense signaling [2]. Among them, salicylic acid and jasmonic acid are two typical phytohormones [3]. Changes in gene expression can be used as molecular markers of defense signaling activation. For example, pathogen-associated molecular patterns (PAMPs) or damage-associated molecular patterns can induce pattern-triggered immunity (PTI) [4,5], which leads to reactive oxygen species bursts and callose deposition [6,7]. There are also effector-triggered immune responses, which elicit hypersensitive reactions, causing cell death [8,9]. It has also been shown that herbivore-related molecular patterns can also trigger plant defense responses [10].

Hemiptera insects, such as aphids, possess piercing–sucking mouthparts. The stylets of aphids can penetrate plant tissue and suck plant fluids [11]. Accompanied by the feeding process, these insects secrete two types of saliva from their salivary glands into plant cells [12]. One is gelling saliva, which has the role of protecting the stylets and assisting feeding, and the other is watery saliva, which has functions such as determining feeding sites and digesting nutrients [13,14,15]. Salivary proteins in saliva can also be involved in the regulation of host plant defense responses. C002 was the first well-characterized salivary effector from the pea aphid *Acyrthosiphon pisum*, which is very important for aphid survival on broad beans [16]. Aphid salivary proteins can trigger defense responses in host plants as well. For example, Mp10 and Mp42 can lead to the chlorosis and cell death of *N. benthamiana* and decrease the fecundity of the green peach aphid *Myzus persicae* [17]. There are also salivary proteins that can promote aphid infestation by inhibiting host plant defense response. For example, MpC002, PIntO1 (Mp1) and PIntO2 (Mp2) of *M. persicae* can inhibit host plant defense responses and significantly increase aphid fecundity [17,18]. Transient overexpression of Mp55 in *Arabidopsis thaliana* reduces the accumulation of a toxic glucosinolate, callose, and hydrogen peroxide (H_2_O_2_), and improves the performance of *M. persicae* [19]. Transient overexpression of the salivary proteins Me10 and Me23 of the potato aphid *Macrosiphum euphorbiae* in *N. benthamiana* significantly increased the fecundity of *M. persicae,* suggesting that they acted as effectors inhibiting plant defense response [20]. In addition, the candidate effectors RpC002 and Rp1 from bird cherry-oat aphid *Rhopalosiphum padi* enhanced aphid host susceptibility by inhibiting the expression of defense-related genes [21]. Salivary proteins of other insects also have the function of modulating host plant defense response. For example, salivary DNase Ⅱ and salivary sheath protein LsSP1 of the small brown planthopper (SBPH) *Laodelphax striatellus* can regulate rice defense response. Rice plants infested by ds*DNase Ⅱ*-treated SBPH accumulated significantly more H_2_O_2_ and callose [22]. The overexpression of LsSP1 attenuated the biosynthesis and response of SA induced by *L. striatellus*, suggesting the potential role of LsSP1 in regulating plant defense in rice [23]. Silencing BtFer1, the salivary ferritin of whitefly *Bemisia tabaci*, enhances the JA-mediated defense signaling pathway and leads to increased callose deposition and proteinase inhibitor production, thereby preventing persistent infestation of whiteflies [24]. In addition, the salivary protein BtFTSP1 of *B. tabaci* significantly inhibited the ferredoxin-mediated defense response in tobacco [25].

The grain aphid *Sitobion avenae* (Fabricius) is one of the most seriously and widely distributed pests of cereal crops, causing damage by directly feeding on phloem sap and transmitting plant viruses [26], such as barley yellow dwarf virus (BYDV), resulting in a decrease in wheat yield and quality [27,28]. Some salivary proteins of wheat aphids have been reported to modulate wheat defense responses. Watery saliva from *S. avenae* induces the expression levels of SA-related genes, enhancing resistance against aphids [29]. A total of 526 putative secreted salivary proteins were identified from the salivary glands of *S. avenae* using transcriptome analysis [30]. Among them, the salivary protein SmCSP4 of *S. avenae* acts as an elicitor that is involved in activating the SA signaling defense pathway of wheat by interacting with TaWRKY76 [31]. The salivary effectors Sm10 and SmC002 enhanced host plant susceptibility and benefited aphid performance [32]. Sm9723 from *S. avenae* and Sg2204 from the greenbug *Schizaphis graminum* were shown to suppress the expression levels of both JA- and SA-responsive genes in wheat plants, thereby promoting aphid fitness on hosts [33,34].

In this study, we screened cyclin-dependent kinase (CDK) from the salivary gland transcriptome sequencing results of *S. avenae*. By overexpressing SaCDK in plants, the roles of SaCDK in modulating wheat defense responses were investigated. Our results showed that SaCDK acted as an effector that significantly impaired plant immunity and promoted host susceptibility. This finding is of important significance not only for uncovering the interaction of cereal aphids and host plants, but also as a potential target of gene knockdown to improve host plant resistance to aphids.

## 2. Results

### 2.1. Sequence Analysis of Candidate Salivary Effector SaCDK

The total length of the *SaCDK* gene contains a 363 bp open reading frame (GenBank accession number: OR838788), encoding 121 amino acids, and the predicted molecular weight is 13.76 kDa. Nine proteins with high homology similarity to SaCDK were selected by BLAST analysis. Amino acid sequence analysis (Figure 1) showed that SaCDK had 91.67% homology similarity with *A. pisum* (GenBank accession number: XP_016659103.1), while the sequence homology with *Sipha flava* (GenBank accession number: XP_025413680.1) was only 52.14%. Phylogenetic analysis indicated that SaCDK was closely related to *A. pisum* and *M. euphorbiae*, clustering into an independent clade (Figure 2).

### 2.2. Spatio-Temporal Expression Profile Analysis of SaCDK

The results of RT-qPCR showed that SaCDK was expressed at different feeding times of wingless adults, and the expression level reached the highest at 24 h after feeding, which was significantly higher than other feeding times (Figure 3A). The expression levels of SaCDK in different instars of *S. avenae* were also different, with the highest expression levels in the first and second instars, and significantly higher than those in the third and fourth instars (Figure 3B).

### 2.3. Transient Expression of SaCDK in N. benthamiana Suppresses Cell Death

Transient overexpression of both mouse pro-apoptotic protein-BAX and *Phytophthora infestans* PAMP-INF1 in tobacco leaves induced programmed cell death (PCD), whereas overexpression of MgCl_2_, pCAMBIA-1300-GFP, which was used as a blank and negative control, did not cause PCD (Figure 4). Meanwhile, overexpression of SaCDK also did not induce PCD symptoms, and significantly inhibited PCD induced by BAX and INF1, which indicated that SaCDK could significantly inhibit the PCD symptoms induced by BAX and INF1.

### 2.4. Delivery of SaCDK Inhibited Callose Deposition in Wheat Leaves

In order to investigate the effect of SaCDK on host plant defense response, SaCDK was cloned into the pEDV6 vector and transferred into wheat via a bacterial type III secretion system (T3SS). As shown in Figure 5A, wheat leaves expressed with AvrRpt2 showed a chlorinated phenomenon, and a large amount of H_2_O_2_ accumulation could be observed after DAB staining, which indicates that *Pseudomonas fluorescens*-mediated T3SS was effective, and effector proteins could be stably expressed and secreted into wheat leaves. Wheat leaves injected with DsRed did not show chlorosis and only a small amount of H_2_O_2_ accumulation. In addition, the infiltration of EtAnH expressing SaCDK did not induce any chlorosis symptoms or H_2_O_2_ accumulation on the wheat leaves.

Aniline blue staining showed that the infiltration of EtAnH expressing DsRed induced significant callose deposition in the infiltrated region of leaves. However, inoculation of the EtAnH strain carrying SaCDK significantly inhibited callose deposition in wheat leaves when compared to the controls (Figure 5B). And as can be seen in Figure 5C, the number of callose deposits in wheat leaves delivered with SaCDK was significantly reduced compared with AvrRpt2 and DsRed.

### 2.5. SaCDK Inhibited the SA and JA Defense Signaling Pathways

To further investigate the roles of SaCDK on modulating wheat defense response, the expression levels of SA and JA signaling pathway defense response genes in wheat leaves were detected at 2 and 4 days post infiltration. As shown in Figure 6A, the expression levels of SA-responsive genes *PR1* and *PAL* in wheat leaves expressing SaCDK were significantly decreased at 2 and 4 days compared with the control group, and the results in Figure 6B showed that the expression levels of JA-associated genes *FAD* and *LOX* genes were also significantly downregulated at 4 days.

### 2.6. SaCDK Enhanced the Survival and Fecundity of S. avenae

*S. avenae* feeding on wheat leaves overexpressing SaCDK resulted in increased survival compared to the control during the recorded three-week period (Figure 7A), and the number of nymphs produced by each aphid on wheat leaves treated with SaCDK was significantly higher than the control.

## 3. Discussion

Salivary proteins secreted by aphids perform an important role in the interactions between aphids and host plants [35,36]. Several salivary proteins of *S. avenae* have been demonstrated to be involved in modulating wheat defense responses. For example, the salivary protein SmCSP4 secreted by *S. avenae* interacts with TaWRKY76 to activate the SA signaling defense pathway in wheat [31]. The salivary effectors Sm10 and SmC002 enhance host plant susceptibility by modulating defense signaling pathways [32]. CDK plays important roles in the coordinated control of cell cycle progression and is used to regulate plant growth and development [37,38]. For example, CDK8 can participate in JA-mediated defense in *Arabidopsis* and regulate the biosynthesis of hydroxycinnamic acid amides, secondary metabolites with defensive activity [39]. This study aimed to deeply investigate the potential roles of a function-uncharacterized salivary protein SaCDK identified from the transcriptome of *S. avenae* salivary glands in regulating wheat defense responses [30]. We found that the expression levels of SaCDK were significantly upregulated after aphid attack in wheat, indicating that this protein may be involved in the modulating the interactions between aphid and wheat. Although the SaCDK sequence does not contain a signal peptide, several studies have shown that salivary proteins without a signal peptide sequence can also be secreted into plants and regulate the defense response of host plants [40,41], and Western blot experiments will be carried out to verify that SaCDK protein can be secreted into plant cells by aphids.

BAX and PAMP-INF1 induced PTI-related PCD [42,43,44]. In this study, our results showed that SaCDK could inhibit PCD induced by BAX/INF1 in *N. benthamiana*. Wheat crops are the main hosts of *S. avenae* [45]; therefore, the role of SaCDK in regulating wheat immunity needs to be further confirmed. *Agrobacterium*-mediated transient expression systems are inefficient in wheat, which seriously hinders the studies on the interaction mechanism between pathogens and wheat plants [46]. Recently, a *P. fluorescens*-mediated bacterial type-III secretion system (T3SS) was successfully utilized to deliver the effector proteins of cereal pathogens into wheat [47,48,49], and a number of salivary proteins of cereal aphids were also identified that stimulated or inhibited the wheat defense response by using a T3SS system [31,32,33,34,50]. Here, we found that wheat leaves overexpressing AvrRpt2 exhibited significant chlorosis and H_2_O_2_ accumulation, which was consistent with previous studies [51,52,53], demonstrating that *P. fluorescens*-mediated T3SS was effective, but no chlorosis phenotype and H_2_O_2_ accumulation were observed in SaCDK-overexpressed wheat leaves, indicating that SaCDK did not cause hypersensitive responses in wheat leaves. Callose accumulation is a typical phenomenon of PTI reaction, which can improve the resistance of host plants to aphids [54,55]. In this study, aniline blue staining showed that overexpression of SaCDK also significantly reduced callose deposition in wheat leaves, suggesting that SaCDK is involved in the inhibition of plant defense.

SA and JA are two important signaling molecules that mediate plant defense response [56,57,58]. A previous study showed that exogenous application of SA and JA significantly reduced plant mortality and the number of the sugarcane aphid *Melanaphis sacchari* in susceptible sorghum genotypes [59]. Some studies have also shown that exogenous application of SA enhances the defense ability of aphid-susceptible wheat varieties against *S. avenae* [60], and that exogenous application of JA inhibits the growth of *M. euphorbiae* population on tomato [61]. Salivary proteins of aphids have been demonstrated to affect plant resistance by regulating SA and JA signaling pathways. Saliva of tomato aphid *Aphis gossypii* activates the SA and JA signal defense pathways, reduces the growth rate of *A. gossypii* population, and thus enhances the resistance of tomato [62]. The salivary protein effector Rp1 of *R. padi* increased the susceptibility of barley by inhibiting the expression of SA and JA defense pathway genes [21]. Transient overexpression of the salivary protein Sm9723 of *S. avenae* and the salivary protein Sg2204 of *S. graminum* reduced the expression of the SA and JA defense-related genes *PAL*, *PR1*, *LOX* and *FAD*, and enhanced the performance of aphids, suggesting that salivary proteins could be involved in the inhibition of plant resistance to aphids [33,34]. We also found that transient overexpression of SaCDK in wheat leaves significantly reduced the expression of the SA signaling pathway key genes *PR1* and *PAL* and the JA signaling pathway key genes *FAD* and *LOX*. Meanwhile, we found that the survival rate and fecundity of *S. avenae* after feeding on wheat leaves infiltrated with SaCDK were significantly improved compared with the control group, which suggests that salivary protein SaCDK enhanced aphid performance by suppressing wheat defense responses. In addition, the effects of *SaCDK* silencing on plant defense and aphid fitness via RNA interference are worthy of further study.

## 4. Materials and Methods

### 4.1. Aphids and Plants

A clone of *S. avenae* was initially collected from a wheat field in Langfang city (39°51′53.21″ N, 116°61′45.96″ E), Hebei Province, northern China. The population was reared on 12-day-old seedlings of aphid-susceptible wheat plants (var. Mingxian 169) under laboratory conditions (16 h light/8 h dark cycle, 20 °C ± 1 °C), and *N. benthamiana* were grown in growth chambers with the following conditions: 23 ± 1 °C with 16 h:8 h (light/dark) photoperiods for four weeks.

### 4.2. Sequence Analysis

The protein molecular weight of SaCDK was predicted by the Compute pI/Mw tool (http://web.expasy.org/com-pute_pi/ (accessed on 27 February 2024 & 3 March 2024)) on Expasy. The multiple alignment of amino acid sequences was performed using Clustal Omega (https://www.Ebi.Ac.uk/Tools/msa/clustalo/ (accessed on 27 February 2024 & 3 March 2024)). A phylogenetic tree was constructed by the neighbor-joining and maximum likelihood methods via MEGA7.0, and 1000 bootstraps were set up to test the sequences. Signal 5.0 (https://services.healthtech.dtu.dk/services/SignalP-5.0/ (accessed on 27 February 2024 & 3 March 2024)) was used for signal peptide prediction.

### 4.3. Transient Expression of SaCDK in N. benthamiana

The full coding sequence of SaCDK was cloned into pCAMBIA1300-GFP, and then pCAMBIA1300-SaCDK-GFP and the empty vector pCAMBIA1300-GFP were transferred into the *A. tumefaciens* GV3101 strain (all primers are listed in Appendix A). The recombinant strain was cultured in LB liquid medium containing kanamycin (50 μg/mL) and rifampicin (20 μg/mL) at 28 °C overnight, and the strains were harvested by centrifugation at 4000× *g* and resuspended to OD_600_ = 0.6–0.8 in buffer solution [10 mmol/L 2-(N-morpholine) ethanesulfonic acid, 20 mmol/L acetosyringone, 10 mmol/L MgCl_2_]. The suspensions were kept in the dark at room temperature for 3 h and soaked into the leaves of *N. benthamiana* (4 weeks old) with a 1 mL syringe without a needle. MgCl_2_, pCAMBIA1300-GFP, pCAMBIA1300-SaCDK-GFP, BAX and INF1 were injected into *N. benthamiana* leaves, and BAX/INF1 was injected at the same location where pCAMBIA1300-SaCDK-GFP was injected 24 h later [17]. The phenotype of tobacco leaves was observed after 4 days of infiltration. Leaves were immersed into the decolorization solution (ethanol/acetic acid = 6:1, *v*/*v*) until complete decolorization and then photographed.

### 4.4. Delivery of SaCDK into Wheat via the Bacterial Type III Secretion System

The SaCDK sequence was cloned into the pEDV6 by gateway recombination (Thermo Fisher Scientific, Waltham, MA, USA) according to the manufacturer’s protocols. Then, the recombinant vector was transferred into the *Pseudomonas fluorescens* strain EtAnH via electroporation [47]. The EtAnH strain carrying the pEDV6: SaCDK recombinant vector was incubated in KB liquid medium containing chloramphenicol (30 μg/mL) and gentamicin (25 μg/mL) for 24 h. The bacterial solution was collected and washed twice with 10 mM MgCl_2_ and resuspended with 10 mM MgCl_2_ to OD_600_ = 1.5–1.8. Cell suspension (OD_600_ = 1.7) was infiltrated into the second leaf of 12-day-old wheat seedlings at the two-leaf stage using a syringe without a needle. Wheat leaves infiltrated with 10 mM MgCl_2_ solution or EtAnH carrying pEDV6: DsRed or pEDV6: AvrRpt2 were used as blank, negative and positive controls, respectively [63]. The infiltrated wheat plants were cultured in a climate chamber at 25 °C, 16 h light/8 h dark photoperiod conditions.

### 4.5. Hydrogen Peroxide Accumulation and Callose Deposition in Wheat Leaves

After 2 days of infiltration, H_2_O_2_ accumulation in wheat leaves was examined using 3′-diaminobenzidine (DAB) staining followed by destaining with decolorization solution (ethanol/acetic acid = 6:1, *v*/*v*) [64,65], and then observed and photographed using Olympus SZX-16 (Olympus Corporation, Tokyo, Japan). Wheat leaves treated with MgCl_2_, DsRed and AvrRpt2 were used as blank, negative and positive controls, respectively. Six biological replicates for each treatment were conducted. Callose deposition was detected with aniline blue according to the histochemical methods described previously [48]. Echo Revolve Hybrid Microscope (Echo Laboratories, San Diego, CA, USA) was used for observation and photography. Fifteen sites were randomly selected on each infiltrated wheat leaf for callose deposits quantity statistics. Six biological replicates were performed for each treatment.

### 4.6. RT-qPCR

Total RNA from aphid *S. avenae* and wheat leaves was extracted using TRIzol reagent (Invitrogen, Waltham, MA, USA) as described previously [30], followed by reverse transcription using the HiScript III 1st Strand cDNA Synthesis Kit (+gDNA wiper) (Vazyme, Nanjing, China). Then, RT-qPCR was used to analyze the relative expression level. Primers were designed using Beacon Designer 7 software and listed in Appendix A. RT-qPCR was performed on an ABI 7500 Real-Time PCR System (Applied Biosystems (Waltham, MA, USA)). The total reaction volume was 20 µL, including 1 µL cDNA, 10 µL 2× Tap Pro Universal SYBR qPCR Master Mix (Vazyme, Nanjing, China), 0.4 µL forward primer and reverse primers, and 8.2 µL ddH_2_O. For the analysis of the Sa4636 spatio-temporal expression profile, thirty adults of *S. avenae* were fed on wheat leaves for 0 h, 6 h, 12 h, 24 h and 48 h, and then collected to characterize the transcript levels of Sa4636 at different feeding stages. Thirty aphids of different nymphal instars and adults were collected to detect the expression of SaCDK. β-actin and NADH of *S. avenae* were selected as internal reference genes.

The expression levels of salicylic acid- and jasmonic acid-related genes, including *PR1*, *PAL*, *FAD* and *LOX* in wheat leaves incubated with SaCDK or DsRed, were examined using RT-qPCR [29], and the housekeeping gene β-actin of wheat was used as the internal reference gene [66]. Three replicates were conducted for each treatment, and each replicate contained three technical replicates. Differential expression was calculated using the 2^–ΔΔCt^ method [67].

### 4.7. Aphid Bioassay

Forty independent biological replicates were performed for each treatment using the DsRed as control. Two days after infiltration of SaCDK and DsRed into wheat leaves, eight adult aphids of *S. avenae* were transplanted into each plastic cage (2.5 × 2.5 × 2.5 cm) clamped to the infiltration area. After 24 h, about five first-instar nymphs remained in each plastic cage. The number of new-born nymphs was recorded and removed after each count every day. The assay was conducted for three weeks, and at the end of the experiment, the number of surviving aphids was recorded. New infiltrated wheat leaves with the same treatment were replaced every four days to ensure continued expression of the target proteins.

### 4.8. Statistical Analysis

All data were analyzed using SPSS Statistics 20.0 software (SPSS Inc., Chicago, IL, USA). The differences among groups were examined using Student’s *t* test, one-way analysis of variance (ANOVA) LSD test and Duncan’s new multiple range test. *p* < 0.05 was considered to be statistically significant.

## 5. Conclusions

Overall, our study suggested that the salivary protein SaCDK of *S. avenae* was shown to be involved in the suppression of plant immunity by inhibiting callose deposition, JA- and SA-associated defense signaling pathways, resulting in a significant enhancement of aphid performance. The results of the current study suggested that SaCDK potentially acts as an effector, playing important roles in suppressing wheat defense.

## Figures and Tables

**Figure 1 ijms-25-04579-f001:**
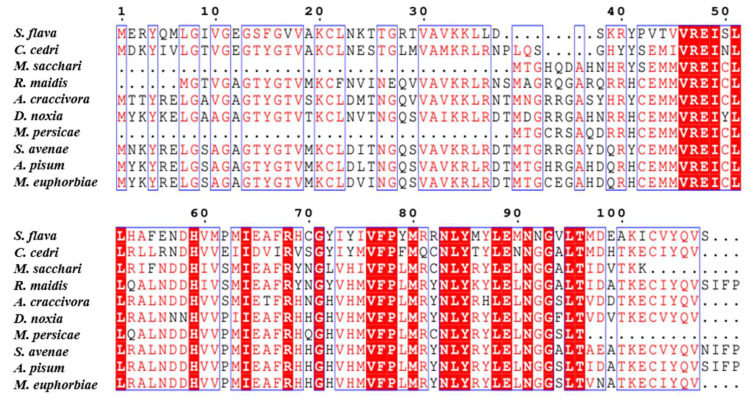
Multiple sequence alignment of cyclin-dependent kinase-like (CDK) and orthologs from other aphid species. The deduced amino acid sequences from nine aphid species include *Acyrthosiphon pisum* (XP_016659103.1), *Macrosiphum euphorbiae* (CAI6360476.1), *Melanaphis sacchari* (XP_025207209.1), *Rhopalosiphum maidis* (XP_026817720.1), *Sipha flava* (XP_025413680.1), *Myzus persicae* (XP_022161581.1), *Cinara cedri* (VVC29270.1), *Aphis craccivora* (KAF0758870.1), and *Diuraphis noxia* (XP_015368308.1). Red shades indicate identical amino acids. Red fonts indicate similar amino acids, and blue boxes include the sequences with identical and similar residues.

**Figure 2 ijms-25-04579-f002:**
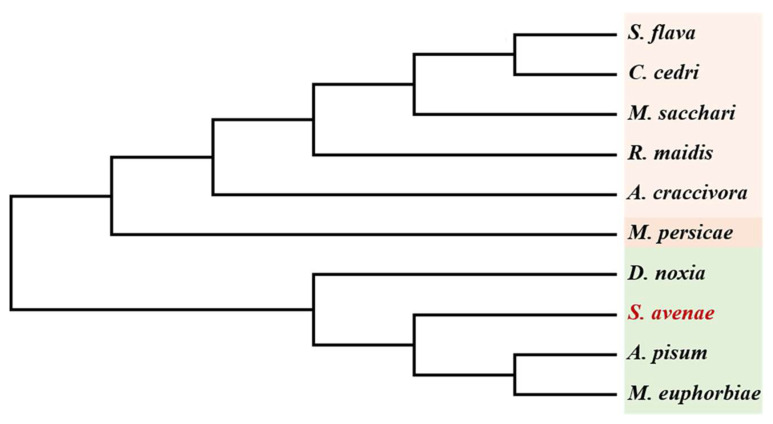
Phylogenetic tree of SaCDK and other homologous aphids was constructed by comparing amino acid sequences. Bootstrap = 1000.

**Figure 3 ijms-25-04579-f003:**
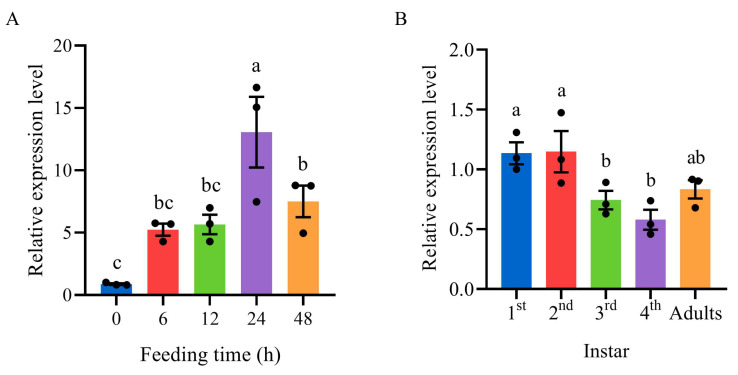
Spatio-temporal expression profile analysis of SaCDK. (**A**) Relative expression levels of SaCDK in apterous adults of *Sitobion avenae* feeding on wheat leaves for different times. The expression level at 0 h was used as the control. (**B**) Relative expression levels of SaCDK in different instars of *S. avenae.* The relative expression of the 1st instar was used as the control. Standard error (SE) is represented by the error bar. Different letters above bars indicate significant difference in the relative expression level (*p* < 0.05, Duncan’s new multiple range test).

**Figure 4 ijms-25-04579-f004:**
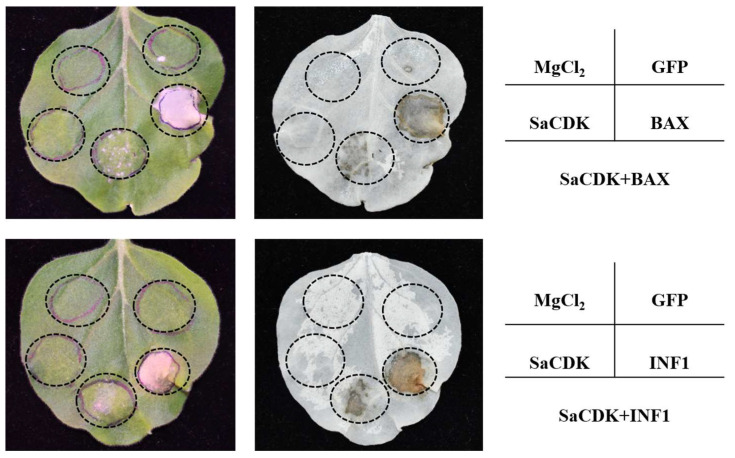
Transient overexpression of SaCDK in *Nicotiana benthamiana* inhibited PCD triggered by BAX and PAMP-INF1. MgCl_2_ and pCAMBIA-1300-GFP were set as blank and negative control groups. Leaves were decolorized using ethanol and acetic acid. Three biological replications were performed for each treatment.

**Figure 5 ijms-25-04579-f005:**
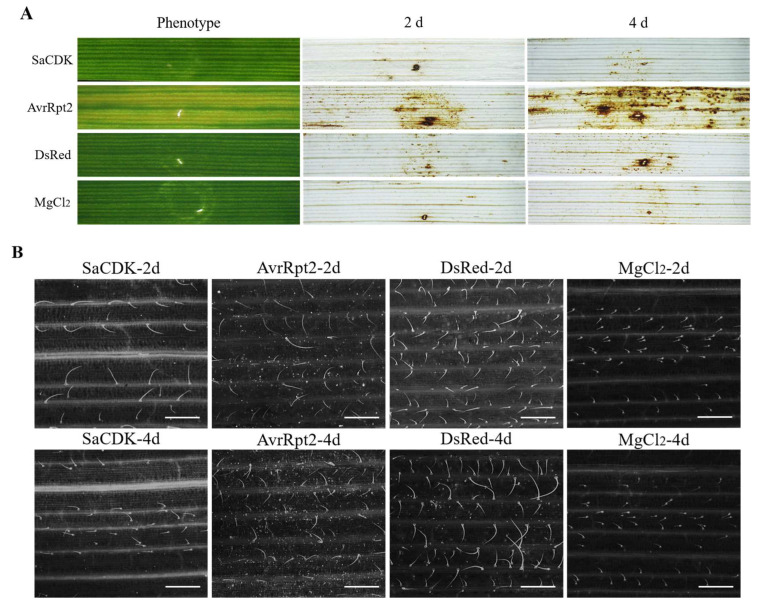
Delivering SaCDK into wheat leaves inhibited callose accumulation. (**A**) Phenotype and Hydrogen peroxide (H_2_O_2_) accumulation in wheat leaves after infiltration with *Pseudomonas fluorescens* EtAnH carrying SaCDK at 2 and 4 days. Leaves infiltrated with MgCl_2_, DsRed or AvrRpt2 were set as blank, negative and positive controls, respectively. (**B**) Aniline blue staining was performed to examine callose deposition in wheat leaves infiltrated with EtAnH carrying SaCDK at 2 and 4 days using epifluorescence microscopy. (**C**) Average number of callose deposits per mm^2^ in wheat leaves inoculated with SaCDK at 2 and 4 days. Wheat leaves treated with MgCl_2_, DsRed or AvrRpt2 were set as controls. Standard error (SE) is represented by the error bar. Different lower-case letters above the bars indicate significant differences between controls and treatments. Bar = 330 μm. Six replicates were conducted for each treatment.

**Figure 6 ijms-25-04579-f006:**
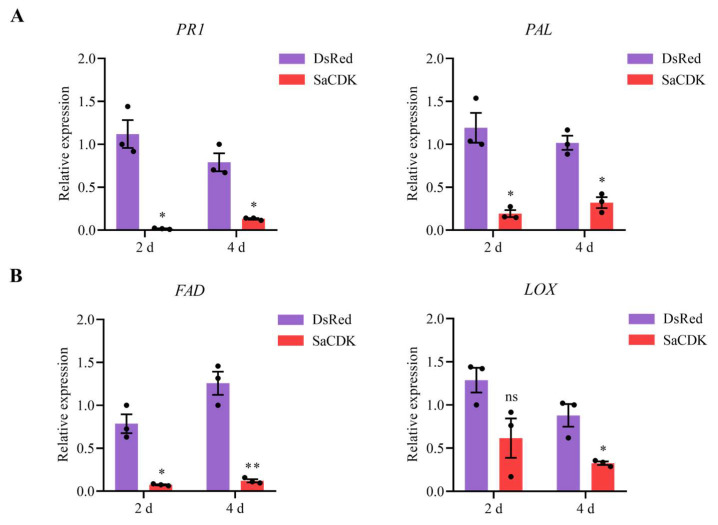
Delivery of SaCDK into wheat leaves suppressed the relative expression of salicylic acid (SA) and jasmonic acid (JA) signaling-related genes. (**A**) Expression levels of pathogenesis-related 1 protein (*PR1*) and phenylalanine ammonia lyase (*PAL*) genes involved in SA signaling pathway. (**B**) Expression levels of Ω-3 fatty acid desaturase (*FAD*) and lipoxygenase (*LOX*) genes associated with JA defense pathway. The values are represented as mean ± SE. Asterisks above the bars indicate significant differences between controls and treatments (Student’s *t* test; * *p* < 0.05; ** *p* < 0.01; ns, not significant).

**Figure 7 ijms-25-04579-f007:**
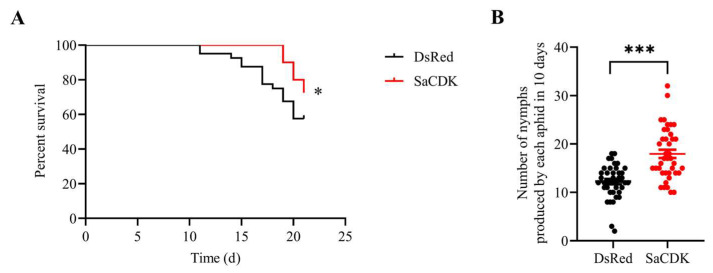
SaCDK promoted *S. avenae* performance. (**A**) Survival rate of *S. avenae* on wheat leaves expressing SaCDK. (**B**) Number of nymphs produced by each *S. avenae* on wheat plants inoculated with SaCDK. Leaves treated with DsRed were set as controls. Each treatment had forty biological replicates (Student’s *t* test; * *p* < 0.05; *** *p* < 0.001). All data are represented as mean ± SE.

## Data Availability

Data are contained within the article and Appendix A.

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
