# Peer review of "Salivary Protein Cyclin-Dependent Kinase-like from Grain Aphid Sitobion avenae Suppresses Wheat Defense Response and Enhances Aphid Adaptation"

_ijms, 2024, doi:10.3390/ijms25094579_

Round 1
Reviewer 1 Report
Comments and Suggestions for Authors
Zhang et al., describe a new salivary effector Sa14636 in aphid Sitobion avenae, and assay its function in insect feeding and wheat defense response. They found Sa14636 can suppress plant cell death triggered by elicitor, and decreased the expression levels of JA and SA-associated genes. Also, they demonstrated that Sa14636 is beneficial to insect. Overall, the experiments were well designed, the results were interesting and will benefit our understanding on insect-plant interaction. I have several concerns before acceptance can be considered.
1) Line 50. I was confused about the function of “determining feeding sites”, can author describe it clearer?
2) Line 51. Delete “And”
3) Line 52-53. Not exactly. Torsten Will (2007, pnas.0703535104) also describe the importance of aphid saliva in insect feeding.
4) Line 68. Incorrect use of comma. Also, I suggested author use the example of salivary proteins with detailed functions in modulating host plant defense. Such as DNase II and LsSP1 in planthopper, the BtFTSP1 in whitefly.
5) I was confused by the name of Sa14636. In NCBI, according to the accession number the authors provided, Sa14636 is the homology of cyclin-dependent kinase. Why author not use the previously annotated name? I strongly suggest the author use cyclin-dependent kinase-like or other similar name, instead of Sa14636. Also, I suggest author briefly introduce cyclin-dependent kinase in Introduction section.
6) In figure 2. Node with low bootstrap values should not be displayed.
7) Figuue 3 and other figures. What is the error bar represented?
8) Line 129. It is better to introduce the BAX and INF1 before the first appearance.
9) Line 130. How did the author conclude the cell death phenotype is “programmed cell death (PCD)”? Any assay supports the PCD?
10) Avoid use word such as “obvious” in scientific article.
11) Line 145. “which indicating”?
12) Line 149. The author should describe clearer. It seems that they directly inject Sa14636 into leaves. To my understanding, it is not.
13) Line 171 “after 2 and 4 days of infiltration”. Not correct. I suggest “at 2 and 4 days post infiltration”.
14) Line 256. The author should tone down their conclusion. According to the present study, we can only conclude that Sa14636 potentially acts as an effector in suppressing wheat defense.
15) Line 270. Other methods, i.e., maximum likelihood method, in constructing phylogenetic tree is necessary.
16) In Method, both 5.3 and 5.7 were associated with RT-qPCR. Can author merge it into one section?
17) Line 5.9. The method used in survival analysis in Fig. 7 should be described.
Comments on the Quality of English LanguageThe manuscript might be benifit from English polish.
Author Response
Dear Reviewer,
We much appreciate your favorite consideration and insight comments on our manuscript of “Salivary protein Sa14636 from grain aphid Sitobion avenae suppress wheat defense response and enhance aphid adaptation” (Manuscript ID: ijms-2927021). We have studied each comment carefully and revised the paper according to your comments. We hope this revision can make our paper more acceptable. The responses to your comments are as following:
- Line 50. I was confused about the function of “determining feeding sites”, can author describe it clearer?
Response: Thanks for the comment. Previous studies have shown that when aphids penetrate a SE with their stylet tips, they start injecting watery saliva into the SE, and, if the penetration is successful, after a period of secretion of watery saliva, the aphid then begins ingesting sap from the SE.
- Line 51. Delete “And”
Response: Thanks for the suggestion. We have revised the manuscript according to your suggestion.
- Line 52-53. Not exactly. Torsten Will (2007, pnas.0703535104) also describe the importance of aphid saliva in insect feeding.
Response: Thanks for the suggestion. The salivary protein C002 of A. pisum is the first well characterized aphid effector, which is essential for continuous phloem feeding and aphid survival. We revised the sentence in Line 53-54 to make it clearer.
- Line 68. Incorrect use of comma. Also, I suggested author use the example of salivary proteins with detailed functions in modulating host plant defense. Such as DNase II and LsSP1 in planthopper, the BtFTSP1 in whitefly.
Response: Thanks for the suggestion. We revised it according to your comments (Line 69-79).
- I was confused by the name of Sa14636. In NCBI, according to the accession number the authors provided, Sa14636 is the homology of cyclin-dependent kinase. Why author not use the previously annotated name? I strongly suggest the author use cyclin-dependent kinase-like or other similar name, instead of Sa14636. Also, I suggest author briefly introduce cyclin-dependent kinase in Introduction section.
Response: Thanks for the suggestion. According to your suggestion, we have revised "Sa14636" to "cyclin-dependent kinase-like (CDK)" in the whole manuscript.
- In figure 2. Node with low bootstrap values should not be displayed.
Response: Revised as suggested.
- Figure 3 and other figures. What is the error bar represented?
Response: Standard error (SE) is represented by the error bar. We provided this information in Figure 3, 5, 7 legends (Line139, 183, 211-212).
- Line 129. It is better to introduce the BAX and INF1 before the first appearance.
Response: Thanks for the suggestion. BAX and INF1 were introduced in Introduction part (Line 142-143).
- Line 130. How did the author conclude the cell death phenotype is “programmed cell death (PCD)”? Any assay supports the PCD?
Response: Thanks for the suggestion. Previous studies shown that mouse pro-apoptotic protein-BAX and Phytophthora infestans PAMP-INF1 could act as elicitors to induce programmed cell death (PDC) in plants (Kamoun et al., 1993; Boumela et al., 2009).
- Avoid use word such as “obvious” in scientific article.
Response: Thanks for the suggestion. We removed it as you suggested (Line143, 159, 247).
- Line 145. “which indicating”?
Response: Sorry for the mistake. We corrected it in the article (Line 160).
- Line 149. The author should describe clearer. It seems that they directly inject Sa14636 into leaves. To my understanding, it is not.
Response: Thanks for the suggestion. We didn’t directly inject Sa14636 into leaves. Sa14636 was actually delivered into wheat leaves using the Pseudomonas fluorescens EtAnH-mediated delivery system.
- Line 171 “after 2 and 4 days of infiltration”. Not correct. I suggest “at 2 and 4 days post infiltration”.
Response: Revised as suggested (Line 189).
- Line 256. The author should tone down their conclusion. According to the present study, we can only conclude that Sa14636 potentially acts as an effector in suppressing wheat defense.
Response: Revised as suggested (Line 389).
- Line 270. Other methods, i.e., maximum likelihood method, in constructing phylogenetic tree is necessary.
Response: Thanks for the suggestion. We added this method in the manuscript (Line 291).
- In Method, both 5.3 and 5.7 were associated with RT-qPCR. Can author merge it into one section?
Response: Thanks for the suggestion. We merged RT-qPCR methods as you suggested (Line 347-357).
- Line 159. The method used in survival analysis in Fig. 7 should be described.
Response: Thanks for the suggestion. The aphid survival was analyzed using Student’s t test, we added it in figure legend of Figure 7 (Line 211).
Reviewer 2 Report
Comments and Suggestions for Authors
Great work overall. Article written in a nice condensed manner, does not get boring or tedious and thus was an enjoyable read. English grammar here and there could use some refining, but not my native language so I am not the one to judge it.
I have only one important issue. The callose deposition experiment with Sa14636 delivery with bacterial secretion needs better elaboration. I am not sure if I (and thus “the reader”), misunderstood it or if it was badly designed. As I understood, you “expressed” virulence factor AvrRpt2, fluorescence protein DsRed and your test protein in wheat separately. Then you compare amounts of callose with negative control. But why, in several places, you say that Sa14636 “reduced” callose deposition? Reduced compared to what? It is false. Sa14636 increased callose deposition compared to negative control. A plant without Sa14636 in it, produced less callose. Sa14636 presence triggers callose deposition, but in much smaller affect compared to DsRed/AvrRpt2. Also, even that might be not true, because you used wrong negative control.
Why didn’t you use a plant injected with P. fluorescens EtAnH with empty expression vector as a negative control. If your negative control is just a MgCl2 solution, how do we know what effect on plant the presence of the bacteria has? Maybe that difference in callose between Sa14636 and MgCl2 is not because Sa14636, but because the P. fluorescens itself triggers small response from plant. Comparing a plant with bacteria and empty vector to a plant with same bacteria and vector but with expressed protein of interest would be much more ideal experiment.
Was it not possible to inject the plant with a mix of prepared bacteria (probably 1:1), ones with AvrRpt2 and others with Sa14636 expressing vectors. Similarly to Sa14636+BAX experiment in N. benthamiana. If the presence of Sa14636 and AvrRpt2 together in the plant would have given the smaller accumulation of callose than just AvrRpt2 alone, then you could say that Sa14636 “reduces” callose deposition.
Why was DsRed chosen to be used in experiment? Is it commonly used in such tests?
If I misunderstood something, then you need to present and explain this test better. If I understood all correctly, then your conclusions from this test need to be much more modest and careful. As it stands now, even the name of 2.4. paragraph seem wrong to me. I don’t see proof that “Sa14636 inhibited H2O2 accumulation”. Why do you assume that there was supposed to be H2O2 accumulation? If you compare Sa14636 to MgCl2, then saying “Sa14636 did not induced significant H2O2 accumulation” is more correct.
Another minor detail is protein 3D structure. It would benefit article if you mentioned/contemplated on it. Just 2-3 sentences in discussion maybe.
How do we know that Sa14636 was fully functional and of correct structure? How does it assemble its 3D configuration? Maybe it needs special modifications done only in insect cells, that don’t happen in bacterial secretion system. Does it simply self-arrange itself into its functional 3D structure spontaneously after secretion? What if it needs specific pH and/or other factors/helper proteins that are only present in insect saliva? Since Sa14636+BAX worked in Nicotiana, it probably assembles itself and works. But imagine that you get a negative result. If Sa14636 would not give a noticeable effect in plant, then how would you know if it did not work because it is not interacting with plant or if it not worked because it was not functional because expression/delivery system?
Other remarks:
Line 19: “suppressed callose deposition in wheat” – I don’t see proof that it suppressed it. I see that, quite opposite, Sa14636 induced callose deposition when compared to negative control.
Line 21: “including” – if you tested many genes, then saying “including these four” (among others) would be correct. As you tested only these 4 and mention all of them here, word “including” is weird English grammar.
Line23: “significant” – was significantly increased.
Lines 40-44: “PAMPs, DAMPs, ROS, ETI, HR, HAMPs” – since these abbreviations are not seen elsewhere in the text and are mentioned only once only in here, I don’t see why are they needed. Full names of these phrases are mentioned and enough.
Line 41: “leading”- leads to?
Line 46: “the stylet can insert plant tissue and suck” – definitely bad grammar.
Line 69: “function modulating” – function of modulating.
Line 73: “causing great decrease yield and quality in wheat” – bad grammar.
Line 85: “we isolated and characterized” – I think you did this in your previous study when you “isolated” candidate gene after transcriptomics. In this study, like you nicely said in abstract, “the function of Sa14636 on wheat defense response and the adaptation of S. avenae was investigated.”
Line 140: 2.4. paragraph name. How you know/prove that Sa14636 inhibits callose? It does not induce callose productions so strongly as compared to virulence protein. But it does induce it compared to MgCl2 solution. You did not deliver both Sa14636+AvrRpt2 together to see if Sa14636 reduces AvrRpt2 effect.
Line 145: “indicating” – which indicates that.
Line 175: “of were also” – grammar.
Line 203-204: if here you talk about your previous transcriptome study, then add citation here. Reference [30] I think.
Line 225-226: Either explain this test better, or this is very wrong conclusion. The sentence about ROS within lines 219-223 (but no obvious chlorosis phenotype and 221 H2O2 accumulation were observed in Sa14636-overexpressed wheat leaves, indicating that 222 Sa14636 did not cause hypersensitive responses in wheat leaves) is good example what I think also represents callose. Unless, the healthy unaffected plant, in normal conditions, is supposed to produce much more callose than it did with Sa14636 presence. But your negative control is MgCl2 solution and it is lower than Sa14636.
Line 268: “adults S. avenae” – adults of S. avenae.
Line 278: “with” – cultured in LB.
Line 315: the 4.7 paragraph about qPCR should be moved upward to 4.3 position (shifting others accordingly). So that order of methods would correspond more the order in which results are presented.
Line 325: “system” – total reaction volume was.
Line 342-343: “Student’ s, Duncan’ s” – the gaps look weird, are they needed?
Line 346: to many “showed” – is involved.
Line 347: callose “inhibition” needs to be explained better or proved.
Comments on the Quality of English Languageremarks included in the review
Author Response
Dear Reviewer,
We much appreciate your favorite consideration and insight comments on our manuscript of “Salivary protein Sa14636 from grain aphid Sitobion avenae suppress wheat defense response and enhance aphid adaptation” (Manuscript ID: ijms-2927021). We have studied each comment carefully and revised the paper according to your comments. We hope this revision can make our paper more acceptable. The responses to your comments are as following:
- I have only one important issue. The callose deposition experiment with Sa14636 delivery with bacterial secretion needs better elaboration. I am not sure if I (and thus “the reader”), misunderstood it or if it was badly designed. As I understood, you “expressed” virulence factor AvrRpt2, fluorescence protein DsRed and your test protein in wheat separately. Then you compare amounts of callose with negative control. But why, in several places, you say that Sa14636 “reduced” callose deposition? Reduced compared to what? It is false. Sa14636 increased callose deposition compared to negative control. A plant without Sa14636 in it, produced less callose. Sa14636 presence triggers callose deposition, but in much smaller affect compared to DsRed/AvrRpt2. Also, even that might be not true, because you used wrong negative control.
Why didn’t you use a plant injected with P. fluorescens EtAnH with empty expression vector as a negative control. If your negative control is just a MgCl2 solution, how do we know what effect on plant the presence of the bacteria has? Maybe that difference in callose between Sa14636 and MgCl2 is not because Sa14636, but because the P. fluorescens itself triggers small response from plant. Comparing a plant with bacteria and empty vector to a plant with same bacteria and vector but with expressed protein of interest would be much more ideal experiment.
Was it not possible to inject the plant with a mix of prepared bacteria (probably 1:1), ones with AvrRpt2 and others with Sa14636 expressing vectors. Similarly to Sa14636+BAX experiment in N. benthamiana. If the presence of Sa14636 and AvrRpt2 together in the plant would have given the smaller accumulation of callose than just AvrRpt2 alone, then you could say that Sa14636 “reduces” callose deposition.
Why was DsRed chosen to be used in experiment? Is it commonly used in such tests?
If I misunderstood something, then you need to present and explain this test better. If I understood all correctly, then your conclusions from this test need to be much more modest and careful. As it stands now, even the name of 2.4. paragraph seem wrong to me. I don’t see proof that “Sa14636 inhibited H2O2 accumulation”. Why do you assume that there was supposed to be H2O2 accumulation? If you compare Sa14636 to MgCl2, then saying “Sa14636 did not induced significant H2O2 accumulation” is more correct.
Response: In this study, as we described in Methods section (Line 335-336), MgCl2 solution is used as blank control as all the prepared bacteria were dissolved in MgCl2 solution before injected into wheat leaves, and the DsRed (red fluorescent protein) was actually used as negative controls in this study (Methods, Line 335-336). As shown in figure 5C, the number of callose deposits were significantly reduced in Sa14636-treated leaves when compared to the DsRed-treated wheat leaves. Therefore, it is concluded that Sa14636 inhibits callose. AvrRpt2 was used as positive control to demonstrate that Pseudomonas fluorescens-mediated T3SS was effective in our study. We can’t prove that Sa14636 inhibited H2O2 accumulation, we are sorry for the mistakes in the title of 2.4 and corrected as “Delivery of SaCDK Inhibited Callose Deposition in Wheat Leaves” (Line 154).
- Another minor detail is protein 3D structure. It would benefit article if you mentioned/contemplated on it. Just 2-3 sentences in discussion maybe.
How do we know that Sa14636 was fully functional and of correct structure? How does it assemble its 3D configuration? Maybe it needs special modifications done only in insect cells, that don’t happen in bacterial secretion system. Does it simply self-arrange itself into its functional 3D structure spontaneously after secretion? What if it needs specific pH and/or other factors/helper proteins that are only present in insect saliva? Since Sa14636+BAX worked in Nicotiana, it probably assembles itself and works. But imagine that you get a negative result. If Sa14636 would not give a noticeable effect in plant, then how would you know if it did not work because it is not interacting with plant or if it not worked because it was not functional because expression/delivery system?
Response: The use of Agrobacterium-mediated transient expression or the bacterial Type III secretion system to introduce proteins into plant cells is currently the most commonly used methods to study the function of insect saliva proteins (Zhang et al., 2022, 2023; Huang et al., 2023). As our results, overexpression of Sa14636 also inhibited the callose deposition and the expression levels of defense genes, suggesting that this protein works and involve in the modulating wheat defense. However, as mentioned by reviewer in this comment, it is difficult to know whether the proteins were fully functional. In the future study, we will investigate the effects of Sa14636-silenced via RNAi to further prove its functions.
References:
- Zhang, Y.; Liu, X.B.; Francis, F.; Xie, H.C.; Fan, J.; Wang, Q.; Liu, H.; Sun, Y.; Chen, J.L. The salivary effector protein Sg2204 in the greenbug Schizaphis graminum suppresses wheat defence and is essential for enabling aphid feeding on host plants. Plant Biotechnol. J. 2022, 20, 2187-2201.
- Zhang, Y.; Fu, Y.; Liu, X.B.; Francis, F., Fan, J., Liu, H., Wang, Q.; Sun, Y.; Zhang, Y.M.; Chen, J.L. SmCSP4 from aphid saliva stimulates salicylic acid-mediated defence responses in wheat by interacting with transcription factor TaWKRY76. Plant Biotechnol. J. 2023, 21, 2389-2407.
- Huang, H.J.; Wang, Y.Z.; Li, L.L.; Lu, H.B.; Lu, J.B.; Wang, X.; Ye, Z.X.; Zhang, Z.L.; He, Y.J.; Lu, G.; Zhuo, J.C.; Mao, Q.Z.; Sun, Z.T.; Chen, J.P.; Li, J.M.; Zhang, C.X. Planthopper salivary sheath protein LsSP1 contributes to manipulation of rice plant defenses. Commun. 2023, 14, 737.
Other remarks:
- Line 19: “suppressed callose deposition in wheat” – I don’t see proof that it suppressed it. I see that, quite opposite, Sa14636 induced callose deposition when compared to negative control.
Response: In this study, MgCl2 solution is used as blank control as all the prepared bacteria were dissolved in MgCl2 solution before injected into wheat leaves, and the DsRed (a red fluorescent protein) was actually used as negative controls in this study (Methods, Line 335-336). As shown in figure 5C, the number of callose deposits were significantly reduced in Sa14636-treated leaves when compared to the DsRed-treated wheat leaves. Therefore, it is concluded that Sa14636 inhibits callose.
- Line 21: “including” – if you tested many genes, then saying “including these four” (among others) would be correct. As you tested only these 4 and mention all of them here, word “including” is weird English grammar.
Response: Revised as suggested (Line 22).
- Line 23: “significant” – was significantly increased.
Response: Revised as suggested (Line 24).
- Lines 40-44: “PAMPs, DAMPs, ROS, ETI, HR, HAMPs” – since these abbreviations are not seen elsewhere in the text and are mentioned only once only in here, I don’t see why are they needed. Full names of these phrases are mentioned and enough.
Response: Revised as suggested (Line 41-45).
- Line 41: “leading”- leads to?
Response: Revised as suggested (Line 42).
- Line 46: “the stylet can insert plant tissue and suck” – definitely bad grammar.
Response: Thanks for the suggestion. We revised this part of the content (Line 47-48).
- Line 69: “function modulating” – function of modulating.
Response: Revised as suggested (Line 69).
- Line 73: “causing great decrease yield and quality in wheat” – bad grammar.
Response: Thanks for the suggestion. We modified this part as “resulting in a decrease in wheat yield and quality” (Line 84-85).
- Line 85: “we isolated and characterized” – I think you did this in your previous study when you “isolated” candidate gene after transcriptomics. In this study, like you nicely said in abstract, “the function of Sa14636 on wheat defense response and the adaptation of avenae was investigated.”
Response: Thanks for the suggestion. We modified it as suggested (Line 96-97).
- Line 140: 2.4. paragraph name. How you know/prove that Sa14636 inhibits callose? It does not induce callose productions so strongly as compared to virulence protein. But it does induce it compared to MgCl2 You did not deliver both Sa14636+AvrRpt2 together to see if Sa14636 reduces AvrRpt2 effect.
Response: Thanks for the suggestion. MgCl2 is used as blank control as all the prepared bacteria were dissolved in MgCl2 solution before injected into wheat leaves, and the DsRed (a red fluorescent protein) was actually used as negative controls in this study (Methods, Line 335-336). As shown in figure 5C, the number of callose deposits were significantly reduced in Sa14636-treated leaves when compared to the DsRed-treated wheat leaves. Therefore, it is concluded that Sa14636 inhibits callose.
- Line 145: “indicating” – which indicates that.
Response: Revised as suggested (Line 160).
- Line 175: “of were also” – grammar.
Response: Revised as suggested (Line 193).
- Line 203-204: if here you talk about your previous transcriptome study, then add citation here. Reference [30] I think.
Response: Revised as suggested (Line 227).
- Line 225-226: Either explain this test better, or this is very wrong conclusion. The sentence about ROS within lines 219-223 (but no obvious chlorosis phenotype and 221 H2O2 accumulation were observed in Sa14636-overexpressed wheat leaves, indicating that 222 Sa14636 did not cause hypersensitive responses in wheat leaves) is good example what I think also represents callose. Unless, the healthy unaffected plant, in normal conditions, is supposed to produce much more callose than it did with Sa14636 presence. But your negative control is MgCl2 solution and it is lower than Sa14636.
Response: Thanks for the suggestion. In this study MgCl2 solution is used as blank control as all the prepared bacteria were dissolved in MgCl2 solution before injected into wheat leaves, and the DsRed (a red fluorescent protein) was actually used as negative controls in this study (Methods, Line 335-336). As shown in figure 5C, the number of callose deposits were significantly reduced in Sa14636-treated leaves when compared to the DsRed-treated wheat leaves. Therefore, it is concluded that Sa14636 inhibits callose.
- Line 268: “adults avenae” – adults of S. avenae.
Response: Revised as suggested (Line 353).
- Line 278: “with” – cultured in LB.
Response: Revised as suggested (Line 305).
- Line 315: the 4.7 paragraph about qPCR should be moved upward to 4.3 position (shifting others accordingly). So that order of methods would correspond more the order in which results are presented.
Response: Revised as suggested (Line 347-357).
- Line 325: “system” – total reaction volume was.
Response: Revised as suggested (Line 350).
- Line 342-343: “Student’ s, Duncan’ s” – the gaps look weird, are they needed?
Response: Revised as suggested (Line 381-382).
- Line 346: to many “showed” – is involved.
Response: Revised as suggested (Line 385).
- Line 347: callose “inhibition” needs to be explained better or proved.
Response: Thanks for the suggestion. the DsRed (a red fluorescent protein) was actually used as negative controls in this study as described in Methods section, Line 335-336). As shown in Figure 5B, C, less callose deposition were observed in wheat leaves treated with Sa14636 when compared to those in the negative control DsRed (a red fluorescent protein). Therefore, we concluded that callose was inhibited.
Reviewer 3 Report
Comments and Suggestions for Authors
Zhang and colleagues present in the manuscript entitled "Salivary protein Sa14636 from grain aphid Sitbion avenae suppress wheat defense response and enhance aphid adaptation" a candidate effector of aphid salivary protein, capable of dampening the plant host response to aphidal feeding. The authors used straight forward methods to discern the role of Sa14636 in surpressing wheat response to aphid feeding.
The study is a sturdy, compact and well conducted study and it is my opinion that it is in the interest of the IJMS audience.
What I see as a minor flaw is the lack of hypothesis, how Sa14636 is mechanically acting to induce the suppression of SA and JA signalling pathways, H2O2 accumulation and callose deposition. If the authors could add a section in the discussion part dealing with its possible cellular role, this would enhance to my feeling the quality of the manuscript. Especially, considering that for A. pisum the ortologous Protein name has been designated as cyclin-dependent kinase-like 1. Maybe cross-checking functions of cyclin-dependent kinases would be a way to discuss this topic.
Minor comments:
L28: Graphical abstract instead of Abstract graphic
Author Response
Dear Reviewer,
We much appreciate your favorite consideration and insight comments on our manuscript of “Salivary protein Sa14636 from grain aphid Sitobion avenae suppress wheat defense response and enhance aphid adaptation” (Manuscript ID: ijms-2927021). We have studied each comment carefully and revised the paper according to your comments. We hope this revision can make our paper more acceptable. The responses to your comments are as following:
- What I see as a minor flaw is the lack of hypothesis, how Sa14636 is mechanically acting to induce the suppression of SA and JA signalling pathways, H2O2 accumulation and callose deposition. If the authors could add a section in the discussion part dealing with its possible cellular role, this would enhance to my feeling the quality of the manuscript. Especially, considering that for pisum the ortologous Protein name has been designated as cyclin-dependent kinase-like 1. Maybe cross-checking functions of cyclin-dependent kinases would be a way to discuss this topic.
Response: Thanks for the suggestion. At present, there are few studies on the function of insect CDK genes. Studies have shown that CDK8 could participate in JA-mediated defense in Arabidopsis and regulate the biosynthesis of hydroxycinnamic acid amides, a secondary metabolite with defensive activity. We have added relevant discussions in the “Discussion” section (Line 220-224). However, the molecular mechanism of its action is still unknown. We have not found relevant literature. In the follow-up study, we will further study its mechanism of action in inhibiting plant defense response by constructing transgenic plants, yeast two-hybrid and other methods.
- L28: Graphical abstract instead of Abstract graphic
Response: Revised as suggested (Line 29).
